# Pterostilbene Attenuates High-Intensity Swimming Exercise-Induced Glucose Absorption Dysfunction Associated with the Inhibition of NLRP3 Inflammasome-Induced IECs Pyroptosis

**DOI:** 10.3390/nu15092036

**Published:** 2023-04-23

**Authors:** Lin Zheng, Pengfei Hou, Jinjin Jing, Min Zhou, Le Wang, Luting Wu, Jundong Zhu, Long Yi, Mantian Mi

**Affiliations:** Research Center for Nutrition and Food Safety, Chongqing Key Laboratory of Nutrition and Food Safety, Institute of Military Preventive Medicine, Third Military Medical University, Chongqing 400038, China; zlinzheng00@163.com (L.Z.); houpengfeilucky@126.com (P.H.);

**Keywords:** pterostilbene, high-intensity swimming exercise, glucose absorption, intestinal epithelium cell, pyroptosis, SIRT3

## Abstract

The study investigated the effect of pterostilbene (PTE) on intestinal glucose absorption and its underlying mechanisms in high-intensity swimming exercise (HISE)-treated mice. Male C57BL/6 mice were treated with PTE for 4 weeks and performed high-intensity swimming training in the last week. Intestinal epithelial cells (IECs) were pretreated with 0.5 and 1.0 μM PTE for 24 h before being incubated in hypoxia/reoxygenation condition. Intestinal glucose absorption was detected by using an oral glucose tolerance test and d-xylose absorption assay, and the levels of factors related to mitochondrial function and pyroptosis were measured via western blot analyses, cell mito stress test, and quantitative real-time polymerase chain reaction. In vivo and in vitro, the results showed that PTE attenuated HISE-induced intestinal glucose absorption dysfunction and pyroptosis in mice intestine. Moreover, PTE inhibited NLRP3 inflammasome and the mitochondrial homeostasis as well as the ROS accumulation in IEC in vitro. Additionally, knockdown of SIRT3, a major regulator of mitochondria function, by siRNA or inhibiting its activity by 3-TYP abolished the effects of PTE on pyroptosis, mitochondrial homeostasis, and ROS generation of IEC in vitro. Our results revealed that PTE could alleviate HISE-induced intestinal glucose absorption dysfunction associated with the inhibition of NLRP3 inflammasome-induced IECs pyroptosis.

## 1. Introduction

Appropriate exercise is beneficial for the prevention of metabolic diseases such as obesity, metabolic syndrome, and cardiovascular disease [1]. However, several studies found that high-intensity exercise (>60% VO_2max_ or 7.2 metabolic equivalent) could induce organ damage, especially the gastrointestinal tract. Metabolic equivalents (MET) are defined as the caloric consumption of an active individual compared with the resting basal metabolic rate, which is 1 MET at rest. VO_2max_ is the maximum (max) rate of oxygen (O_2_) consumption one’s body is able to utilize during exercise done at maximal intensity whilst breathing air at sea level. On average, an individual utilizes 3.5 mL of O_2_ per kg of body weight per minute (mL/kg/min). Therefore, one MET equals a VO_2_ of 3.5 mL/kg/min [2]. Carbohydrates, including glucose, were mainly absorbed in the small intestine, which was mediated by the sodium-dependent glucose co-transporter 1 (SGLT1) and glucose transporter 2 (GLUT2) in intestinal epithelial cells (IEC) [3]. Running either with 70% VO_2max_ substantial exposure or with 30% and 50% VO_2max_ in thermos-neutral ambient conditions resulted in remarkable reductions in active and passive carbohydrate absorption [4]. As for the mechanism, the circulatory–gastrointestinal pathway involving redistribution of blood flow to working muscle from peripheral circulation played a vital role in the exercise-induced intestinal injury [5]. The study found that the portal blood flow was decreased by 20% within 10 min, and by 80% after 1 h of running at 70% VO_2max_, which would induce the intestinal ischemia-reperfusion (I/R) injury [6]. Nevertheless, the cellular and molecular mechanisms, as well as the protective strategy of high-intensity exercise-induced intestinal glucose absorption dysfunction, were still not clear.

Our previous study showed that excessive exercise led to a significant mitochondrial dysfunction of IECs and intestine inflammation [7]. Increasing evidences suggest pyroptosis, with cell swelling, chromatin condensation, and cell membrane rupture as significant features, plays a key role in immune and inflammatory diseases of the intestine, and it is a new caspase-1-dependent inflammatory modality of cell death as well [8]. The activated caspase-1 triggers pyroptosis by maturating pro-inflammatory cytokines such as interleukin-1β (IL-1β) and interleukin-18 (IL-18), as well as cleaving Gasdermin D (GSDMD) [9]. Moreover, the activation of caspase-1 relies on the inflammasome, for instance, node-like receptor (NLR) family, NLR pyrin domain-containing protein 3 (NLRP3), which is the adaptor of apoptosis-associated speck-like protein containing a CARD domain (ASC) and procaspase-1 [10]. As for the NLRP3 inflammasome, it could be activated by diverse triggers, including extracellular ATP, lipopolysaccharide, mitochondrial DNA (mtDNA), and reactive oxygen species (ROS), and subsequently result in damages in the cell [11,12]. It is well known that the mitochondrial homeostasis is a complex process maintained by multiple proteins, such as dynamin-related protein 1 (Drp1), mitofusin 1/2 (Mfn1/2) and respiratory complex activities [13,14]. Meanwhile, many studies have confirmed that ROS generation and release induced by mitochondrial dysfunction and dynamics disorder are pivotal for programmed cell death and act as a key player in IEC fate decisions and in coordinating cellular metabolism, immunity, and stress responses [15]. On the basis of our previous findings, we supposed that high-intensity exercise-induced intestinal glucose absorption dysfunction might be associated with pyroptosis.

Pterostilbene (PTE), a natural polyphenol, is proved to be a great anti-oxidant and anti-inflammatory agent [16,17]. Our previous study demonstrated that PTE could ameliorate mitochondrial dysfunction through AMP-activated protein kinase/Silent information regulator1 (SIRT1)/Proliferator-activated receptor coactivator 1α (PGC-1α) pathway [18]. However, thus far, almost no study reported the effects of PTE on high-intensity exercise-induced intestinal glucose absorption dysfunction. Silent information regulator3 (SIRT3), which belongs to nicotinamide adenine dinucleotide (NAD^+^)-dependent deacetylase family, is mainly localized in mitochondria. A previous study found that SIRT3 deficiency led to an increased ROS level [19]. On the other hand, our previous study found that mitochondrial SIRT3 enrichment could promote mitochondrial function and decrease ROS generation [20]. In the present study, we aimed to investigate the effect of PTE supplementation on the high-intensity exercise-induced intestinal glucose absorption dysfunction and its underlying molecular mechanisms.

## 2. Materials and Methods

### 2.1. Animals and Experimental Protocol

Fifty-two male C57BL/6 mice (8 weeks, 18–22 g) were obtained from the Experimental Animal Centre of the Third Military Medical University (Chongqing, China). They were provided food and water ad libitum and housed under controlled conditions of temperature (22 ± 2 °C), humidity (55 ± 5%), and 12 h light/dark cycle. After one week of acclimatization, the mice were randomly divided into four groups (*n* = 13 each group): the sedentary control group (Sed), exercise training group (Ex), exercise training combined with 50 mg/kg·bw PTE treatment group (Ex + PTE50), and 100 mg/kg·bw PTE treatment group (Ex + PTE100). PTE (HPLC ≥ 98%, A0752, Must Bio-Technology, Chengdu, China) was dissolved in normal saline and was given by gavage once a day for four weeks at a dose of 50 mg/kg·bw or 100 mg/kg·bw, while the mice in the Sed group and Ex group were treated with an equal volume of normal saline as a vehicle. The dosage was chosen according to the previous study, in which, it was shown that these doses of PTE demonstrated its beneficial effects on exercise endurance [18,21].

After accepting PTE administration for two weeks, mice accessed the adaptive swimming exercise for one week and high-intensity swimming exercise (HISE) for another one week following the protocol, with a slight modification of the previous study [22,23]. Briefly, mice were placed into four plastic boxes (90 × 50 × 40 cm) filled with 30 ± 1 cm deep water (25 ± 1 °C). All mice except the mice in the Sed group were initially adapted to the water environment for 5 days: day 1, two periods of 30 s swimming with a 2-h interval between the swimming periods; day 2, two periods of 2 min of swimming with a 2-h interval; day 3, three periods of 10 min of swimming with a 5-min interval; day 4, two periods of 15 min of swimming with a 5-min interval; and day 5, a period of 30 min with no pause. From the beginning of the fourth week, the above mice were subjected to swimming for 2 h with no pause from day 1 to day 5. Meanwhile, the mice in the Sed group were exposed to shallow water, which allowed them to stand and keep their heads out of the water without swimming. After swimming, all mice were gently dried with a cloth towel and a hair dryer. The mice swimming for 2 h once a day was equivalent to 60–75% VO_2max_ according to the previous study [24]. The body weight was measured every week and the food intake was measured every two days.

After HISE training, an oral glucose tolerance test (OGTT) and d-xylose absorption assay were performed, and then all mice were fasted overnight and sacrificed. Serum was prepared by centrifugation (4 °C, 3900 rpm, 15 min) and then stored at −80 °C for further analyses. The small intestine tissues were isolated and washed with PBS. Parts of the small intestine tissues were fixed with 4% paraformaldehyde for immunofluorescence microscopy analysis, and the rest were frozen immediately in liquid nitrogen, and kept at −80 °C for further analyses.

All animal experiments were approved by the Institutional Animal Care and Use Committees of the Third Military Medical University (Chongqing, China; Approval SYXC-2017-0002) and followed the National Research Council Guidelines.

### 2.2. Oral Glucose Tolerance Test (OGTT) and d-Xylose Absorption Assay

OGTT was performed twice at pre-experiment and post-experiment after the mice had fasted for 12 h. Blood was sampled from the tail vein before (*t* = 0) and 15, 30 min after oral load of glucose (2 g/kg body weight). Blood glucose levels were measured by handled glucometer (OneTouch Ultra Easy, Johnson & Johnson, New Brunswick, NJ, USA).

On the other day, mice fasted for 12 h and were then given a 3% solution of d-xylose dissolved in normal saline by gavage. After urinary samples were collected, the urine d-xylose level was assayed using a d-xylose ELISA Kit (Mlbio, Shanghai, China).

### 2.3. Caspase-1 p20 Activity Assay

Caspase-1 p20 activity was quantified by using the corresponding ELISA Kit (Mlbio, Shanghai, China) according to the manufacturer’s instructions.

### 2.4. Immunofluorescence Staining

Fresh small intestines were fixed with 4% paraformaldehyde at 4 °C for 24 h, dehydrated, paraffin-embedded, and sectioned (5 μm) for immunofluorescence staining. After being incubated with 0.5% Triton X-100 for 1 h, sections were then blocked with 5% BSA for 30 min and incubated at 4 °C overnight with primary antibodies against GLUT2 (1:1000, Abcam, Cambridge, UK) and SGLT1 (1:1000, Abcam, UK). After being washed with PBS, sections were incubated with fluorescent-labeled secondary antibodies (Alexa Fluor 488-labeled Goat Anti-Rabbit IgG, Cy3 labeled Goat Anti-Mouse IgG, Beyotime, Shanghai, China) at 37 °C for 2 h and were then incubated with 4′,6-diamidino-2-phenylindole (Beyotime, Shanghai, China). Subsequently, they were washed with PBS and mounted with Fluoromount-G (SouthernBiotech, Birmingham, AL, USA). The fluorescence was visualized using laser scanning fluorescence microscopy (Leica, Wetzlar, Germany). The mean fluorescence intensity was measured using Image-J V1.8.0.112 software.

### 2.5. Mitochondrial Respiratory Complex Activities Assay

The activities of mitochondrial respiratory chain complexes I, II, III, IV, and V in mitochondria isolated from the mouse small intestine samples were analyzed with the corresponding kits (Solarbio, Beijing, China) according to the manufacturer’s instructions.

### 2.6. Analysis of Mitochondrial DNA (mtDNA) Content

The mtDNA was isolated from the small intestines using a ONE-4-ALL DNA Mini-Preps Kit (Sangon Biotech, Shanghai, China). Relative amounts of mtDNA were measured by comparing their amplification products with those of β-actin and NADH dehydrogenase subunit 1. The primer sequences are shown in Table 1.

### 2.7. Cell Culture and Treatment

Intestinal epithelial cells (IEC-6) were cultured in DMEM (Invitrogen, Waltham, MA, USA) supplemented with 10% of fetal bovine serum (HyClone, Logan, UT, USA), 1% glutamine (Beyotime, Shanghai, China), and 1% penicillin-streptomycin (Beyotime, Shanghai, China) in a 5% CO_2_ humidified incubator at 37 °C. The hypoxia/reoxygenation (H/R) model in vitro was established as previously described [25]. To simulate hypoxic conditions, IEC-6 cells were incubated in a microaerophilic system (Thermo, Waltham, MA, USA) with 5% CO_2_ and 1% O_2_ and balanced with 94% N_2_ for 4 h. Then, IEC-6 cells were cultured for 3 h under normoxic conditions to achieve reoxygenation. Cells from the 3rd to 6th passages were used for the following experiments. When the cells grew to 80–90% confluence, they were pretreated with PTE (0.5 or 1 μM) (HPLC purity ≥ 98%, Must Bio-Technology, Chengdu, China) or DMSO (as the control; Sigma, Cleveland, OH, USA) at the indicated concentrations for 24 h before being exposed to H/R. To study the involvements of SIRT3 in the PTE-mediated protective effect against H/R, the selective SIRT3 inhibitor 3-TYP (MCE, Wuhan, China) and SIRT3 siRNA (Thermo Fisher, Waltham, MA, USA) were used to treat IEC-6 cells before PTE treatment.

### 2.8. Small Interference RNA (siRNA) Transfection

For RNA interference experiments, IEC-6 cells were transfected with SIRT3 siRNA or control siRNA (Thermo Fisher, Waltham, MA, USA) using Lipofectamine 2000 transfection reagent (Invitrogen, Waltham, MA, USA) mix in Opti-MEM reduced serum medium (Invitrogen, Waltham, MA, USA) following the manufacturer’s instructions. Subsequently, cells were treated with PTE or DMSO as previously described, and harvested for further experiments.

### 2.9. Cell Viability Assay

Cell Counting Kit-8 (CCK-8) (Dojindo, Kumamoto, Japan) was used to measure cell viability. Briefly, IEC-6 cells (8 × 10^3^ cells per well) were seeded into 96-well plates and treated with PTE in different concentrations (0, 1, 5, 10, 15, 20 μM) and different time periods (0, 12, 24, 36, 48 h). At the end of treatment, CCK-8 (10 μL/well) was added to each well at 37 °C for 2 h. Cell viability was detected by absorbance measurements using a monochromator microplate reader (Molecular Devices, San Jose, CA, USA) at 450 nm. Cell viability was calculated from the ratio of the optical density of the experimental cells to that of the control cells (set as 100%).

### 2.10. Glucose Uptake Assay

A 2-deoxy-2-[(7-nitro-2,1,3-benzoxadiazol-4-yl) amino]-d-glucose (2-NBDG) glucose uptake assay kit (Biovision, New Castle, DE, USA) was used to measure glucose uptake of IEC-6 cells following the manufacturer’s instructions. In brief, IEC-6 cells (4 × 10^4^ cells per well) were seeded in a 24-well plate and exposed to the different PTE treatments. Subsequently, the medium was carefully removed without disturbing the cells, and then, glucose uptake mix was gently added to each well to incubate cells at 37 °C with 5% CO_2_ for 30 min. Then, cells were harvested, washed once with ice-cold 1× analysis buffer, and resuspended in 400 µL of 1× analysis buffer. The fluorescence intensity was detected by the monochromator microplate reader (Molecular Devices, San Jose, CA, USA).

### 2.11. Intracellular ROS Measurement

The production of intracellular ROS was measured using 2′,7′-dichlorofluorescein diacetate (DCFH-DA, Beyotime, Shanghai, China) following the manufacturer’s instructions. After being exposed to the various PTE treatments, the IEC-6 cells were loaded with DCFH-DA (10 μM) at 37 °C for 30 min, then washed three times by PBS. Finally, the fluorescence intensity was detected by the monochromator microplate reader (Molecular Devices, San Jose, CA, USA) with an emission wavelength of 525 nm and an excitation wavelength of 488 nm.

### 2.12. ELISA

Serum parameters, including intestinal fatty acid binding protein (I-FABP), lactate dehydrogenase (LDH), IL-1, and IL-18 were quantified by using the corresponding ELISA kit (Mlbio, Shanghai, China) according to the manufacturer’s instructions.

After the IEC-6 cells were exposed to the indicated treatments, the culture supernatants were harvested. Subsequently, the lactate dehydrogenase (LDH) and IL-1β release levels were determined using the corresponding ELISA kit (Mlbio, Shanghai, China) according to the manufacturer’s instructions.

### 2.13. Seahorse XFp Cell Mito Stress Test

Cell mito stress test was measured using XFp analyzer (Seahorse Bioscience, Agilent Technologies, Santa Clara, CA, USA). Briefly, IEC-6 cells (1 × 10^4^ cells per well) were seeded in XF cell culture microplates. After treatment, IEC-6 cells were cultured in a 37 °C non-CO_2_ incubator for 1 h at 37 °C with assay medium containing 10 mM glucose, 1 mM pyruvate, and 2 mM glutamine, which were adjusted to pH 7.4. Subsequently, Oligomycin, carbonyl cyanide-p-trifluoromethoxyphenylhydrazone, and rotenone/antimycin from cell mito stress test kit (Seahorse Bioscience, Agilent Technologies, Santa Clara, CA, USA) was injected into a hydrated sensor cartridge for analyzing. The parameters, including basal respiration, ATP production, maximal respiration, proton leak, and non-mitochondrial oxygen consumption, were tested according to the manufacturer’s instructions.

### 2.14. RNA Extraction and Quantitative Real-Time Polymerase Chain Reaction (qRT-PCR)

Total RNA was extracted from the small intestines and IEC-6 cells using the Trizol reagent (Invitrogen, Waltham, MA, USA) according to the manufacturer’s instructions. Reverse transcription of mRNA into cDNA was carried out with PrimeScript RT master mix (Takara, Beijing, China) and qRT-PCR was carried out with qTOWER 2.2 (Analytik Jena, Jena, Germany) using TB Green^®^ Premix Ex Taq™ II (Takara, Dalian, China) according to the manufacturer’s instructions. Gene-specific primers were designed by Sangon Biotech (Shanghai, China). Each sample was processed in triplicate and normalized to β-actin with the 2^−ΔΔCT^ method. The gene-specific primer sequences were listed in Table 1.

### 2.15. Western Blotting

The protein samples were extracted from the small intestines and IEC-6 cells by RIPA buffer added with a protease/phosphatase inhibitor cocktail (Roche Applied Science, Indianapolis, IN, USA).

The protein samples were electrophoretically separated using 12–15% SDS-PAGE and transferred to PVDF membranes (Bio-Rad, Hercules, CA, USA). The membranes were blocked for 1 h with 5% dried skimmed milk at room temperature, incubated overnight at 4 °C with primary antibodies listed in Table 2, and then incubated for 1 h with the corresponding secondary antibodies (peroxidase-conjugated goat anti-rabbit igg; peroxidase-conjugated goat anti-mouse IgG, ZSGB-Bio, Wuhan, China) at room temperature. Finally, the proteins were visualized using a chemoluminescence system (Fusion, Paris, France) with Millipore Immobilon ECL substrate (Millipore, Burlington, MA, USA). Densitometry analysis was conducted by Image J V1.8.0.112 software.

### 2.16. Statistical Analysis

Data analysis was performed with GraphPad prism 6.0 (GraphPad Software, Inc., La Jolla, CA, USA). All experimental data were expressed as mean ± SEM. Differences among groups were determined with one-way ANOVA followed by Tukey’s test. *p*-values less than 0.05 were considered statistically significant. All experiments were repeated at least three times.

## 3. Results

### 3.1. PTE Attenuated HISE-Induced Glucose Absorption Dysfunction in Mice

Compared with the mice in the the Sed group, HISE with or without PTE treatment resulted in a moderate reduction in body weight, while there was no significant difference in food intake (Figure 1A,B). The results of OGTT and d-xylose absorption assay revealed that urinary d-xylose and blood glucose levels of the mice in the Ex-group were significantly lower than those in the mice of the Sed group (Figure 1C–E). However, the PTE administration in either 50 mg/kg·bw/day or 100 mg/kg·bw/day for four weeks significantly increased the level of d-xylose and blood glucose (Figure 1C–E). Furthermore, the expression levels of GLUT2 and SGLT1 in the small intestine assayed by immunofluorescence in the mice of the Ex-group were remarkably decreased compared to those in the mice of the Sed group, which were significantly attenuated by the PTE supplementation (Figure 1F,G). In addition, the mRNA and protein levels of GLUT2 and SGLT1 were noticeably reduced in the mice of the Ex group, which was reversed by PTE treatment (Figure 1H–J). The results indicated that PTE administration could significantly attenuate the HISE-induced intestinal glucose absorption dysfunction.

### 3.2. PTE Weakened Pyroptosis in the Small Intestine of Mice Accessing to HISE

The question of whether the pyroptosis of IECs was involved in the process was explored. As expected, the significant enhancement of plasma I-FABP, LDH, IL-1β, and IL-18 levels were observed in the mice in the Ex-group compared to the mice in the Sed group, but PTE administration remarkably removed the influence of HISE (Figure 2A–D). In addition, the mRNA expression levels of gsdmd, IL-1β, IL-18 (Figure 2F) and the protein levels of IL-1β, IL-18 (Figure 2G–H) were higher in the small intestine of the Ex-group, while PTE supplementation reduced this increase. Furthermore, the expression of NLRP3 with caspase-1 p20 in mRNA and protein levels (Figure 2G–H), and the activation of caspase-1 p20, were significantly increased in the Ex-group, which was attenuated by PTE administration (Figure 2E). These results suggested that absorption dysfunction in HISE-treated mice may be associated with pyroptosis, which could be improved by PTE administration.

### 3.3. PTE Eliminated Intestinal Mitochondrial Dysfunction in HISE Treated Mice

Next, as shown in Figure 3A–E, PTE administration could rescue HISE-induced decreases in the mitochondrial respiratory chain (MRC) complexes activities. The amount of mtDNA was significantly decreased in the mice of Ex group, which were remarkably increased by PTE administration (Figure 3F). Furthermore, HISE treatment led to the mitochondrial fission/fusion disorder by significantly increasing the mRNA and protein expression levels of Drp1 and Fis1 and reducing those levels of Mfn2 and Opa1 (Figure 3G–I). Collectively, these results demonstrated that HISE were able to induce mitochondrial dysfunction, including the decreased MRC complexes activities and mitochondrial fission/fusion disorder. However, PTE supplement notably improved the mitochondrial fission/fusion disorder (Figure 3G–I). In addition, the mRNA and protein levels of SIRT3 were significantly lowered in the mice in the Ex-group but were increased in the mice with PTE administration. Thus, PTE were supposed to ameliorate the mitochondrial dysfunction associated with the regulation of SIRT3.

### 3.4. PTE Attenuated Glucose Absorption Dysfunction and Pyroptosis in H/R Treated IEC-6 Cells

The H/R model of IEC-6 cells was established in vitro to mimic the splanchnic ischemia-hypoxia condition induced by HISE in vivo. PTE administration at a dose of 0.5 and 1.0 μM for 24 h significantly attenuated the decreased cell viability that H/R induced (Figure 4A). As shown in Figure 4B, the 2-NBDG level decreased in the H/R group, which was notably ameliorated by PTE treatment. In addition, H/R significantly reduced the expression of GLUT2 and SGLT1 both at the mRNA and protein levels in IEC-6 cells, while PTE treatment obviously increased these levels (Figure 4C–E). Furthermore, consistent with in vivo results, H/R also induced higher LDH and IL-1β release and enhanced the protein expression of NLRP3 and Caspase-1 p20, which was remarkably reversed by PTE administration (Figure 4F–I). Taken together, these results demonstrated that H/R could induce glucose absorption dysfunction and pyroptosis in IEC-6 cells, and these impairments could be effectively attenuated by PTE treatment.

### 3.5. PTE Improved H/R-Induced Mitochondrial Dysfunction in IEC-6 Cells

The ROS levels were significantly increased in H/R-treated IEC-6 cells but were notably suppressed by PTE administration (Figure 5A). As shown in Figure 5B–F, H/R induced significant reductions in basal respiration, ATP production, and maximal respiration, but an increase in proton leak, which implied that H/R induced mitochondrial respiration capacity injury. Interestingly, the PTE administration remarkably reversed the compromised mitochondrial respiration capacity. Additionally, H/R severely enhanced the protein expression levels of Drp1 and Fis1 but decreased Mfn2 and Opa1 in IEC-6 cells, which were remarkably reversed by the PTE administration (Figure 5G,H). Overall, these results indicated that PTE could improve H/R-induced mitochondrial dysfunction and the ROS accumulation of IEC-6 cells.

### 3.6. PTE Suppressed H/R-Induced Glucose Absorption Dysfunction and Pyroptosis as Well as Mitochondrial Dysfunction of IEC-6 Cells through a SIRT3-Dependent Mechanism In Vitro

In an in vivo study, we found the expression of SIRT3 was obviously decreased in the small intestine of the mice in response to HISE, but was increased with PTE administration. To further verify the role of SIRT3 in H/R-induced IEC-6 cells in vitro, 3-TYP and SIRT3 siRNA were respectively used to inhibit the activity of SIRT3 and knockdown expression of SIRT3. The results showed the effect of PTE administration attenuated the decreased cell viability and 2-NBDG level induced by H/R (Figure 6A–D), as well as the increased protein levels of NLRP3 and Caspase-1 p20 induced by H/R were abolished in the presence of 3-TYP or SIRT3 siRNA (Figure 6E–H). In short, these results suggested that the protective effects of PTE on H/R-induced glucose absorption dysfunction and pyroptosis in IEC-6 cells were regulated by SIRT3 in vitro.

Furthermore, PTE administration also reversed the H/R-induced mitochondrial fission/fusion disorder, as evidenced by abnormal expression of Drp1, Fis1, Mfn2, and Opa1, while pretreatment with the SIRT3 inhibitor 3-TYP or SIRT3 siRNA significantly impaired PTE-induced mitochondrial dynamics recovery in vitro (Figure 6I–L). Moreover, as shown in Figure 6M,N, PTE administration notably ameliorated the increased ROS level induced by H/R, but this effect was inhibited by 3-TYP or SIRT3 siRNA. Thus, the above results suggested the protective effects of PTE in H/R-induced mitochondrial dysfunction and pyroptosis of IEC-6 cells in a SIRT3-dependent mechanism.

## 4. Discussion

During prolonged exercise, approximately 30–50% of athletes experience glucose absorption dysfunction, which is a vital cause for the impairment of exercise performance and repair processes [3,26]. Yet there have been few studies focusing on the cellular and molecular mechanism as well as the protective strategy. This study found high-intensity swimming exercise of mouse and H/R of IEC-6 cells in vitro could decrease the glucose absorption and increase NLRP3 inflammasomes-induced pyroptosis as well as mitochondrial dysfunction in intestines or IECs. Supplementation of PTE significantly improved the high-intensity swimming-induced intestinal glucose absorption dysfunction and pyroptosis. The mechanism indirectly demonstrated that the protective effects of PTE were a SIRT3-dependent mechanism in vitro.

The IEC morphology integrity and competent function of nutrient transporters on the intestinal enterocytes were important for achieving nutrient intake requirements during prolonged exercise [27]. To the contrary, gastrointestinal dysfunction impaired glucose absorption and the endurance performance of athletes [28]. In our study, after HISE of mice, plasma I-FABP level, as an intestinal injury marker, significantly increased and the expression of SGLT1, and GLUT2 decreased obviously, which is consistent with the previous study [5]. Interestingly, we found PTE supplementation could significantly improve the HISE-induced intestinal glucose absorption dysfunction and reverse intestinal injury. In brief, this study provided the experimental evidence that PTE could be considered as a novel dietary strategy for HISE-induced intestinal glucose absorption.

It is well known that pyroptpsis is an inflammasome-mediated inflammatory programmed cell death [29]. Inflammasomes are multiprotein cytoplasmic complexes that have the ability to recognize various pathogen-associated molecular patterns and damage-associated molecular patterns, as well as ROS [30]. Subsequently, NLRP3 can connect pro-caspase-1 through ASC, ultimately forming the inflammasome to activate caspase-1 [10]. Consequently, caspase-1 p20, the activated form of caspase-1, could induce the generation of mature IL-1β and IL-18 [31]. As for the mechanism of HISE-induced intestinal glucose absorption and the protective effect of PTE, our research demonstrated that HISE increased the expression of caspase-1 p20, IL-1β, and IL-18 in the small intestinal tissues, which was consistent with the results of the IEC-6 cells in H/R in vitro. For the first time, our findings provided evidence of the mechanism that both HISE-induced intestinal absorption dysfunction and the protective effect of PTE were associated with pyroptosis. Additionally, except for pyroptosis, whether other cellular mechanisms such as apoptosis and necrosis participate in the HISE-induced IEC injury need to be elucidated in the future.

As we all know, mitochondria homeostasis is important for signal transduction and cell energy metabolism. Moreover, mitochondria contribute to ROS generation and the maintenance of cellular homeostasis. In addition, various mechanisms are involved in the activation of the NLRP3 inflammasome, including the generation of ROS, urate crystals, ATP, silica, potassium efflux, and bacterial pore-forming toxins, and the release of lysosomal cathepsins [11,32,33]. Recently, ROS generation was demonstrated to be critical for NLRP3 activation, as supported by the evidence that inhibiting the ROS obviously suppressed NLRP3 inflammasome activation and inflammatory responses in vivo and in vitro [34,35]. Moreover, the current study provided new evidence that H/R could induce a significant increase of ROS, which could be eliminated by PTE treatment. Importantly, mitochondrial fission and fusion played critical roles in maintaining functional mitochondria when cells experience metabolic or environmental stresses. Fis1 and Drp1 are core components of the mitochondrial fission machinery, while the mitochondrial fusion process is controlled by Mfn1/2 and Opa1 to a large extent [36]. According to the reports, in response to the high levels of cellular stress, mitochondrial fission contributes to remove the damaged mitochondria and facilitate cell death. Moreover, mitochondrial Mfn2 may also be involved in massive ROS production. Additionally, specific knock-down of Mfn2 brought about aberrant ROS aggregation in mouse embryo fibroblasts [37]. Analogously, Mfn2 deletion increased hydrogen peroxide content in skeletal muscle and liver [38]. In our in vivo and in vitro research, the expression levels of Drp1 and Fis1 were severely increased in the mice with HISE and in the H/R IEC-6 cells, while Mfn2 and Opa1 were significantly decreased, which implied that HISE may induce the mitochondrial fission but inhibit fusion. Interestingly, we found that PTE supplementation could effectively reverse these injuries in HISE-treated mice. Furthermore, the Mfn2 downregulation was likely connected with altered mitochondrial respiration complex I and III, which prompted excessive ROS production [39]. In these studies, the enzymatic activities of MRC complexes were decreased in HISE-treated mice but were increased by PTE supplementation. These findings indirectly suggested that mitochondrial dysfunction may play a vital role in HISE-induced pyroptosis of intestinal epithelial cells. Additionally, H/R induced pyroptosis of IEC-6 may be due to the excessive ROS generation, which could be improved by PTE supplementation.

Nutritional supplements have been proven to be effective for the prevention of exercise-induced gastrointestinal inflammation [40]. PTE, as a natural demethylated analog of resveratrol (RSV), possesses great health benefits, including anti-inflammation, anti-oxidation, and anti-aging, which are similar to those of RSV [16,17]. In this study, we found that PTE supplementation could attenuate HISE-induced intestinal glucose absorption dysfunction and IEC pyroptosis as well as mitochondrial dysfunction. Recent studies have revealed that SIRT3 is involved in regulating mitochondrial homeostasis [41,42]. For example, SIRT3 deficiency led to a reduced intracellular ATP level and increased ROS level [19]. Meanwhile, we previously found that mitochondrial SIRT3 enrichment could promote mitochondrial function and decrease ROS generation [20]. Additionally, RSV could inhibit NLRP3 inflammasome activation by preserving mitochondrial integrity through activating SIRT3 [43]. Our study showed that PTE treatment significantly increased basal respiration, ATP production, maximal respiration, and improved proton leak induced by H/R. In addition, the expression of SIRT3 was decreased by HISE but increased by PTE supplementation in vivo and in vitro. Meanwhile, our studies also showed that PTE suppressed H/R-induced glucose absorption dysfunction and pyroptosis as well as mitochondrial dysfunction of IEC-6 cells in a SIRT3-dependent mechanism in vitro. However, whether PTE-attenuated intestinal injuries could be abolished by the SIRT3 deficiency in the mouse HISE model should be further investigated by infecting viral vector-delivered SIRT3 siRNA in the mouse GI system or performing the SIRT3^−/−^ mouse model.

In conclusion, this study provides a novel effect of PTE that it could significantly alleviate intestinal glucose absorption in HISE-treated mice. As for the mechanism, it is associated with the inhibition of NLRP3 inflammasome-induced IECs pyroptosis and mitochondrial dysfunction. SIRT3 may be the molecular target of the improvement in PTE-suppressed H/R-induced glucose absorption dysfunction and pyroptosis as well as mitochondrial dysfunction of IEC-6 cells in vitro. These results elucidated the potential protective effects and underlying mechanisms of PTE on HISE-induced intestinal glucose absorption dysfunction.

## Figures and Tables

**Figure 1 nutrients-15-02036-f001:**
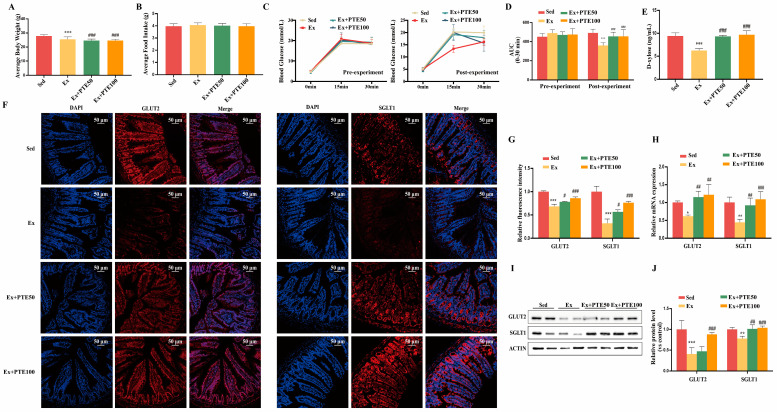
PTE attenuated HISE-induced glucose absorption dysfunction in mice. (**A**) Body weight of mice in different groups. (**B**) Food intake of mice in different groups. (**C**). Time course of blood glucose levels after oral load of glucose (2 g/kg body weight). (**D**) The area under the curve (AUC) values of plasma glucose level. (**E**) Urine level of d-xylose. (**F**,**G**) Representative images of GLUT2 and SGLT1 immunofluorescence staining in small intestinal tissues (Scale bars: 50 μm) and quantification displayed by bar graph. (**H**) The mRNA expression levels of glut2 and sglt1 in small intestine tissues. (**I**) The protein levels of GLUT2 and SGLT1. (**J**) Quantification of protein expression levels of GLUT2 and SGLT1. Data were expressed as means ± SEM. * *p* < 0.05, ** *p* < 0.01, *** *p* < 0.001 versus Sed group; ^#^
*p* < 0.05, ^##^ *p* < 0.01, ^###^ *p* < 0.001 versus the Ex group. Multiple groups were tested by one-way ANOVA followed by Tukey’s test.

**Figure 2 nutrients-15-02036-f002:**
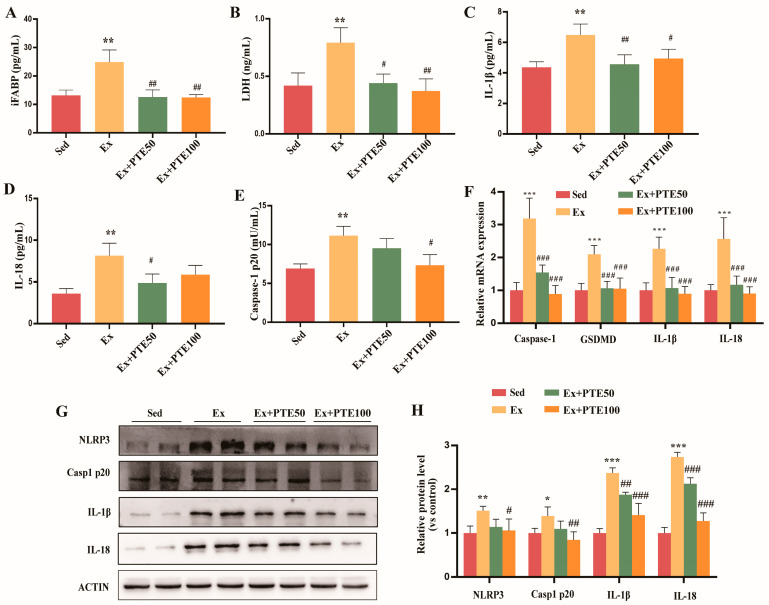
PTE weakened pyroptosis in the small intestine of mice accessing HISE. (**A**–**D**) The plasma I-FABP, LDH, IL-1β, and IL-18 level. (**E**) Caspase-1 p20 activity. (**F**) The mRNA expression levels of caspase-1, gsdmd, IL-1β, and IL-18 in small intestine tissues. (**G**) The protein levels of NLRP3, Cleaved caspase-1 (Casp1 p20), IL-1β and IL-18. (**H**) Quantification of protein expression levels of NLRP3, Cleaved caspase-1 (Casp1 p20), IL-1β, and IL-18. Data were expressed as means ± SEM. * *p* < 0.05, ** *p* < 0.01, *** *p* < 0.001 versus the Sed group; ^#^ *p* < 0.05, ^##^ *p* < 0.01, ^###^ *p* < 0.001 versus the Ex group. Multiple groups were tested by one-way ANOVA followed by Tukey’s test.

**Figure 3 nutrients-15-02036-f003:**
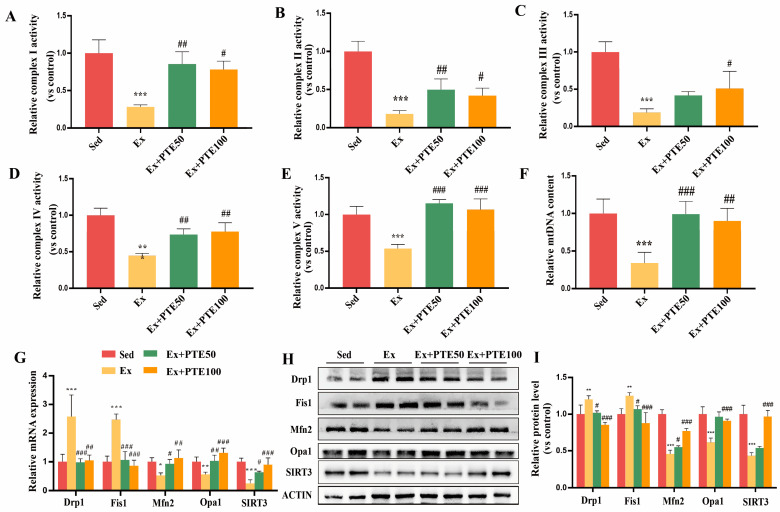
PTE eliminated mitochondrial dysfunction in HISE-treated mice. (**A**–**E**) The activities of mitochondrial respiratory chain complex I, II, III, IV, and V. (**F**) The expression of the mtDNA content. (**G**) The mRNA expression levels of drp1, fis1, mfn2, opa1, and sirt3 in small intestine tissues. (**H**) The protein levels of Drip1, Fis1, Mfn2, Opa1, and SIRT3. (**I**) Quantification of protein expression levels of Drip1, Fis1, Mfn2, Opa1, and SIRT3. Data were expressed as means ± SEM. * *p* < 0.05, ** *p* < 0.01, *** *p* < 0.001 versus Sed group; ^#^
*p* < 0.05, ^##^
*p* < 0.01, ^###^
*p* < 0.001 versus the Ex group. Multiple groups were tested by one-way ANOVA followed by Tukey’s test.

**Figure 4 nutrients-15-02036-f004:**
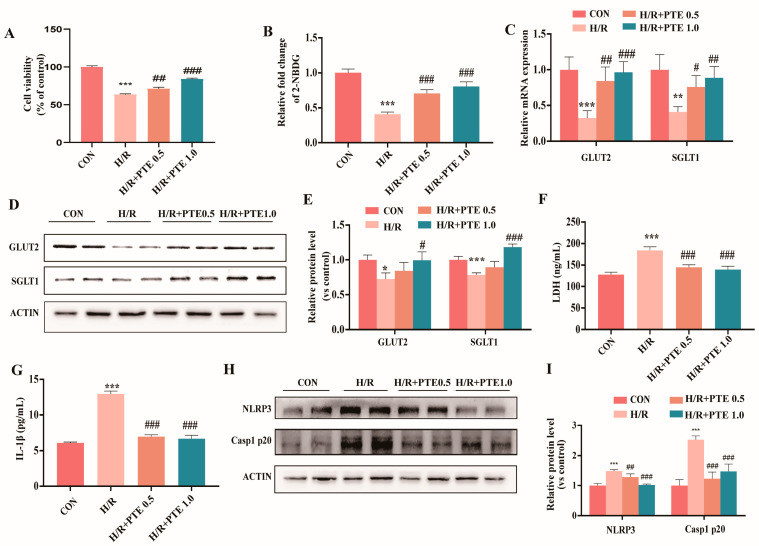
PTE attenuated glucose absorption dysfunction and pyroptosis in H/R-treated IEC-6 cells. (**A**) Cell viability of IEC-6 cells. The cells were treated with PTE at a dose of 0.5 and 1.0 μM for 24 h. Cell viability was calculated from the ratio of the optical density of the experimental cells to that of the control cells (set as 100%); (**B**) Fluorescence intensity change of 2-NBDG. (**C**) The mRNA expression levels of glut2 and sglt1 in IEC-6. (**D**) The protein levels of GLUT2 and SGLT1. (**E**) Quantification of protein expression levels of GLUT2 and SGLT1. (**F**,**G**) LDH and IL-1β release. (**H**) The protein levels of NLRP3 and Caspase-1 p20. (**I**) Quantification of protein expression levels of NLRP3 and Caspase-1 p20. Data were expressed as means ± SEM. * *p* < 0.05, ** *p* < 0.01, *** *p* < 0.001 versus CON group; ^#^
*p* < 0.05, ^##^
*p* < 0.01, ^###^ *p* < 0.001 versus H/R group. Multiple groups were tested by one-way ANOVA followed by Tukey’s test.

**Figure 5 nutrients-15-02036-f005:**
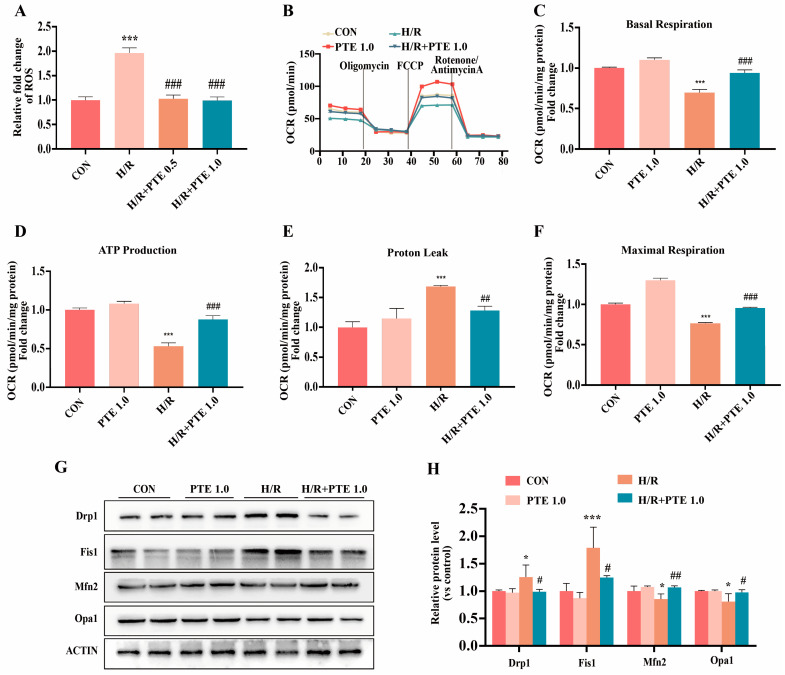
PTE improved H/R-induced mitochondrial dysfunction in IEC-6 cells. (**A**) Fluorescence intensity of ROS in cells. (**B**) Oxygen consumption rates of IEC-6 cells. (**C**) Basal respiration. (**D**) ATP production. (**E**) Proton leak. (**F**) Maximal respiration. (**G**) The protein levels of Drip1, Fis1, Mfn2, and Opa1. (**H**) Quantification of protein expression levels of Drip1, Fis1, Mfn2, and Opa1. Data were expressed as means ± SEM. * *p* < 0.05, *** *p* < 0.001 versus CON group; ^#^ *p* < 0.05, ^##^ *p* < 0.01, ^###^ *p* < 0.001 versus H/R group. Multiple groups were tested by one-way ANOVA followed by Tukey’s test.

**Figure 6 nutrients-15-02036-f006:**
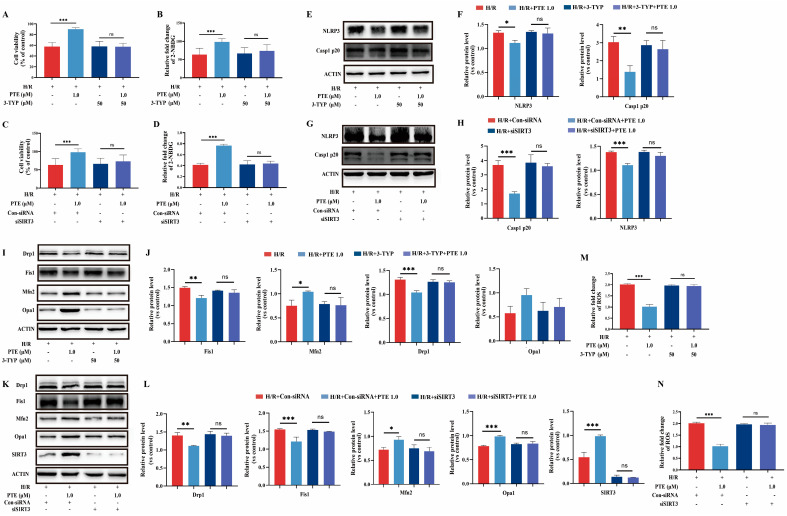
PTE suppressed H/R-induced glucose absorption dysfunction and pyroptosis as well as mitochondrial dysfunction of IEC-6 cells through a SIRT3-dependent mechanism in vitro. IEC-6 cells were pretreated with 3-TYP for 1 h or transfected with control siRNA or SIRT3 siRNA for 24 h, then treated with PTE before being exposed to H/R. (**A**,**B**) Cell viability was calculated from the ratio of the optical density of the experimental cells to that of the control cells (set as 100%). (**C**,**D**) Relative fluorescence intensity of 2-NBDG in cells. (**E**–**H**) The protein levels and quantification of NLRP3 and Caspase-1 p20. (**I**–**L**) The protein levels and quantification of Drip1, Fis1, Mfn2, and Opa1. (**M**,**N**) Relative fluorescence intensity of ROS in cells. Data were expressed as means ± SEM. * *p* < 0.05, ** *p* < 0.01, *** *p* < 0.001, ns: *p* > 0.05. Multiple groups were tested by one-way ANOVA followed by Tukey’s test.

**Table 1 nutrients-15-02036-t001:** Primers used for qRT-PCR.

Genes	Species	Forward Primer (5′→3′)	Reverse Primer (5′→3′)
Glut2	mouse	CTCAGCTTTATTCTGGGCAATC	TTTCTGGACAGAAGAGCAGTAG
Sglt1	mouse	GCCATGTTTTCCACTAATCGTG	CACTTCCAATGTTACTGGCAAA
Glut2	rat	CACCAGCACATACGACACCAGAC	TGGACACAGACAGAGACCAGAGC
Sglt1	rat	CTATCAGCGTCGTCACCGTCTTG	GGCTCCTCCTCTCCTGCATCC
Gsdmd	mouse	CTAGCTAAGGCTCTGGAGACAA	GATTCTTTTCATCCCAGCAGTC
Casp1	mouse	AGAGGATTTCTTAACGGATGCA	TCACAAGACCAGGCATATTCTT
IL-1β	mouse	GCAGCAGCACATCAACAAGAGC	AGGTCCACGGGAAAGACACAGG
IL-18	mouse	AGACCTGGAATCAGACAACTTT	TCAGTCATATCCTCGAACACAG
Sirt3	mouse	TCTATACACAGAACATCGACGG	GCATGTAGCTGTTACAAAGGTC
Drp1	mouse	ACTGATTCAATCCGTGATGAGT	GTAACCTATTCAGGGTCCTAGC
Fis1	mouse	CCTGGTTCGAAGCAAATACAAT	CTTTTCATATTCCTTGAGCCGG
Mfn2	mouse	TCTCCCTCTGACACCTGCCAAC	ACACCACTCCTCCGACCACAAG
Opa1	mouse	CTTACATGCAGAATCCTAACGC	CCAAGTCTGTAACAATACTGCG
Nd1	mouse	TCCCCTACCAATACCACACCC	ATTGTTTGGGCTACGGCTCG
β-actin	mouse	CTACCTCATGAAGATCCTGACC	CACAGCTTCTCTTTGATGTCAC
β-actin	rat	CTGAGAGGGAAATCGTGCGTGAC	AGGAAGAGGATGCGGCAGTGG

β-actin, actin, beta; Casp1, caspase 1; Drp1, dynamin 1-like; Fis1, fission, mitochondrial 1; Gsdmd, gasder. Min D; IL-1β, interleukin 1 beta; IL-18, interleukin 18; Mfn2, mitofusin 2; Nd1, NADH dehydrogenase subunit 1; Opa1, OPA1, mitochondrial dynamin like GTPase; Sirt3, sirtuin 3; Glut2, solute carrier family 2 (facilitated glucose transporter); Sglt1, solute carrier family 5 (sodium/glucose cotransporter).

**Table 2 nutrients-15-02036-t002:** Antibodies used for Western blot analyses.

Antibody	Company	Dilution
GLUT2	Abcam	1:1000
SGLT1	Abcam	1:1000
NLRP3	Abcam	1:1000
CASPASE-1 P20	Proteintech	1:1000
IL-1β	Abcam	1:1000
IL-18	Abcam	1:1000
DRP1	BD biosciences	1:1000
FIS1	BD biosciences	1:1000
MFN2	Abcam	1:1000
OPA1	BD biosciences	1:1000
SIRT3	Cell signaling	1:1000
β-ACTIN	Cell signaling	1:1000

β-actin, actin, beta; Casp1, caspase 1; Drp1, dynamin 1-like; Fis1, fission, mitochondrial 1; IL-1β, interleukin 1 beta; IL-18, interleukin 18; Mfn2, mitofusin 2; Opa1, OPA1, mitochondrial dynamin like GTPase; NLRP3, NLR family, pyrin domain containing 3; Sirt3, sirtuin 3; Glut2, solute carrier family 2 (facilitated glucose transporter); Sglt1, solute carrier family 5 (sodium/glucose cotransporter).

## Data Availability

The data presented in this study are available in article. Further information can be required from the corresponding author if necessary.

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
