# Peer review of "Pterostilbene Attenuates High-Intensity Swimming Exercise-Induced Glucose Absorption Dysfunction Associated with the Inhibition of NLRP3 Inflammasome-Induced IECs Pyroptosis"

_nutrients, 2023, doi:10.3390/nu15092036_

Round 1

Reviewer 1 Report

This article tried to study the effect of pterostilbene (PTE) on intestinal glucose absorption and its underlying mechanisms in high-intensity exercise (HIE) treated mice. By using high-intensity swimming training on mice for the last week after PTE oral gavage for 4 weeks or treating intestinal epithelial cells (IECs) with hypoxia/reoxygenation condition after PTE uptake for 24 h, measuring intestinal glucose absorption, levels of factors related to mitochondrial function, NLRP3 inflammasome-induced pyroptosis and ROS accumulation, the authors concluded that PTE could alleviate HIE-induced intestinal glucose absorption dysfunction associated with the inhibition of NLRP3 inflammasome-induced IECs pyroptosis by activating SIRT3. My comments and suggestions are as follows:

1. Overall the manuscript has well-designed experiments and consistent results in confirming what was occurring in mice occurred in cellular models. However, the authors failed to confirm the role of SIRT3 in mediating the PTE-attenuated HIE-induced glucose absorption dysfunction associated with the inhibition of NLRP3 inflammasome-triggered IEC pyroptosis (Fig. 6) in the mouse HIE model. They should try to inoculate or infect viral vector-delivered control siRNA or SIRT3 siRNA in the mouse GI system or at least treat mice with 3-TYP to see whether the PTE-attenuated intestinal injuries could be abolished in the mouse HIE model. 

2. In the first paragraph of the Introduction, the authors should explain what do VO2max and metabolic equivalent mean for the first time the terms were used so that readers not in the field understand well.

3. Although the manuscript writing is logical and fluent, there are problems with English grammar. This manuscript should be thoughtfully edited by professional and native English speakers.

4. What was the reason (rationale) for using swimming exercises to do HIE? Why did they not let mice run on running wheel or treadmill-based apparatus that are more commonly used for mouse exercise experiments?

5. Are there any differences in the measurements tested in this manuscript when comparing mice swimming exercise with running on a running wheel?

Reviewer 2 Report

The authors evaluated the effect of pterostilbene in modulating the inflammatory and death process.

1-The WB for GLUT2 (Fig.1I) does not clearly show how pterostilbene restores GLUT2 expression to SED levels. Expression appears to be reduced equally in all EX with and without PTE. Do you have a more representative WB? The restore of the sedentary condition is not clear 

2-In the protein characterization of the NLRP3-inflammasome (Fig.2G) you have reported only the full length form of GSDM-D but the active 30kDA form responsible for pyroptosis is not present. If you want to affirm the activation of intestinal pyroptosis following exercise (EX-group) you should show the active form of gasdermine.

3-It would be appropriate to add some data with respect to the cytotoxic effects of PTE on the curve (concentration/time) (Fig.4A). A flow cytometric evaluation of PI and cell cycle can better characterize its effects on cell death, not only CCK8 which gives informations only related to the proliferative status of the cells. 

4-In the representative images of cell viability to demonstrate how PTE restores hypoxic damage following H/R (hypoxia/reoxygenation) (FIG.6A-C), the IEC-6 control without H/R damage should also be shown to demonstrate good basal cell status. In this way it will be possible to state that the H/R and the reduction of SIRT3 actually reduce the viability of the cells and the PTE abolishes the induced insult.

Round 2

Reviewer 1 Report

1. I don’t see why not SIRT3-/- mice are suitable for the HIE animal model. The authors could adjust the same affordable speed, distance, and duration time for mice to run on a treadmill running machine (mice should be forced to perform the same level of HIE).

2. Since the conclusion for the role of SIRT3 was purely established in cellular models but HIE is an in vivo term, the authors should change the title and not use HIE (maybe use hypoxia/reoxygenation-induced glucose absorption dysfunction, instead) to better match the demonstrated results if they decide not to perform the SIRT3-/- mouse model or the mouse model suggested by me. Alternatively, they should remove “by activating SIRT3” from the title.

3. Since there is a possibility that high-intensity swimming and running could be different and this paper only focused on swimming, the authors should specifically mention high-intensity swimming but not generally high-intensity exercise for the experiment description.
